# Study on Sensor Fault-Tolerant Control for Central Air-Conditioning Systems Using Bayesian Inference with Data Increments

**DOI:** 10.3390/s24041150

**Published:** 2024-02-09

**Authors:** Guannan Li, Chongchong Wang, Lamei Liu, Xi Fang, Wei Kuang, Chenglong Xiong

**Affiliations:** 1School of Urban Construction, Wuhan University of Science and Technology, Wuhan 430065, China; leegna@163.com (G.L.); wangchongchong0927@163.com (C.W.); 18120230851@163.com (W.K.); xcl0012@163.com (C.X.); 2Anhui Province Key Laboratory of Intelligent Building and Building Energy-Saving, Anhui Jianzhu University, Hefei 230601, China; 3Key Laboratory of Low-Grade Energy Utilization Technologies and Systems (Chongqing University), Ministry of Education of China, Chongqing University, Chongqing 400044, China; 4State Key Laboratory of Green Building in Western China, Xi’an University of Architecture & Technology, Xi’an 710055, China; 5College of Civil Engineering, Hunan University, Changsha 410082, China; xfang@hnu.edu.cn

**Keywords:** heating, ventilation, and air-conditioning (HVAC), sensors, fault-tolerant control, data increments, multiple linear regression–Bayesian inference

## Abstract

A lack of available information on heating, ventilation, and air-conditioning (HVAC) systems can affect the performance of data-driven fault-tolerant control (FTC) models. This study proposed an in situ selective incremental calibration (ISIC) strategy. Faults were introduced into the indoor air ($T_{tz1}$) thermostat and supply air temperature ($T_{sa}$) and chilled water supply air temperature ($T_{chws}$) sensors of a central air-conditioning system. The changes in the system performance after FTC were evaluated. Then, we considered the effects of the data quality, data volume, and variable number on the FTC results. For the $T_{tz1}$ thermostat and $T_{sa}$ sensor, the system energy consumption was reduced by 2.98% and 3.72% with ISIC, respectively, and the predicted percentage dissatisfaction was reduced by 0.67% and 0.63%, respectively. Better FTC results were obtained using ISIC when the $T_{tz1}$ thermostat had low noise, a 7-day data volume, or sufficient variables and when the $T_{sa}$ and $T_{chws}$ sensors had low noise, a 14-day data volume, or limited variables.

## 1. Introduction

### 1.1. Background

Statistics show that building energy consumption has exceeded one-third of the total global energy consumption [1,2,3], and 60% is consumed by building energy systems (for ventilation, heating, and cooling). With the development of the economy and the improvement in living standards, people have a higher and higher demand for indoor air environments [4,5]. The application of HVAC systems in buildings is also becoming increasingly widespread [6,7,8]. Many data-driven techniques have been proposed in order to improve the performance of HVAC systems [9]. These techniques and applications include system fault detection and diagnosis [10,11,12,13], and building energy consumption prediction [14,15], control, and optimization [16,17]. The key to these applications and technologies lies in the building sensing environment. The within-control-loop sensors in building systems have a huge impact on energy consumption [18], control efficiency, indoor environmental quality, and comfort [19,20]. Sensor faults can negatively impact data-driven applications [21] and big data analytics [22], which, in turn, diminishes the value of smart buildings. Regardless of how advanced data-driven applications are, the sensing environment remains a core prerequisite for realizing smart buildings [23,24].

### 1.2. Research and Challenges in Fault-Tolerant Control of Building Energy Systems

The performance of complex systems designed using conventional feedback control may degrade or become unstable when a component in the system has a fault [25]. Therefore, new methods of constructing control systems need to be developed to address these issues [26,27]. Furthermore, the expected performance and stability of a system should be maintained while allowing for component faults. Systems that are tolerant to soft faults are called fault-tolerant control (FTC) systems and are widely used in industries such as aerospace, chemicals, and nuclear power. In HVAC systems, FTC systems are able to automatically withstand the loss of individual components, keeping the system stable and working at tolerable levels. Wang et al. [28] applied autoencoder virtual in situ calibration to a solar photovoltaic–thermal heat pump system. The in situ calibration method can reduce the sensor system error by over 95%. Yoon [29] proposed a virtual in situ calibration method that couples Bayesian inference with an autoencoder. This method completely eliminates sensor errors and energy waste. Zhao et al. [30] combined a virtual sensor in situ calibration method with Gaussian mixture model clustering to realize the online diagnosis and real-time repair of sensors in air-handling units. Li et al. [31] used an autoencoder to establish a sensor model for a virtual sensor in situ calibration method. Then, they discussed the calibration performance for a single fault and concurrent faults, with sensor accuracies of more than 90.75% and 88.5%, respectively, and the energy efficiency was increased by 21% and 17% for the water pump and the supply fan, respectively. For a series of fault scenarios in air-handling units, Liu et al. [32] proposed a novel fault detection, diagnosis, and self-calibration method based on Bayesian inference. The method showed effective performance with a small number of datasets. To address the problem of virtual sensor errors having a negative effect on the in situ calibration accuracy of the corresponding physical sensors, Koo et al. [33] proposed and adopted a simultaneous in situ calibration method. Li et al. [34] proposed a general-regression-improved Bayesian inference calibration method with a calibration accuracy of over 97%. This method improved the accuracy by 2.8–20% compared with the energy conservation Bayesian inference and principal component analysis methods. Li et al. [35] applied a Bayesian-inference-based method to a practical building energy system and demonstrated calibration accuracy and efficiency improvement using this strategy. Tian et al. [36] proposed a sensor drifting bias calibration method based on Bayesian inference and an autoencoder. It realized an optimal control strategy for data center cooling supply systems. The simulation results showed that the calibration accuracies were more than 92.60% and 96.34% for the scenarios of individual-sensor and multiple-sensor drifting bias, respectively, which filled a gap in the research.

Despite the high calibration accuracy of these advanced FTC methods, satisfactory results are highly dependent on having sufficient HVAC data for modeling. For HVAC systems that do not collect data on a regular basis due to time-consuming data collection processes and expensive measurement sensors, the available historical data are not sufficient for modeling FTC methods. To address this issue, it is necessary to explore a real-time FTC model that can solve or at least mitigate the lack of historical training data.

### 1.3. Research and Challenges in Data Incremental Learning for FTC

In real-world scenarios, long-term continuous data collection is important for FTC models before and after a model’s implementation. For every data-driven FTC method, a certain amount of historical data is required for training and validation. Although building energy consumers and managers want to implement FTC method modeling as soon as possible, the available historical data may not always be sufficient. Data incremental learning (DIL) [37,38,39] utilizes continuously collected data to extend the knowledge of an existing model (i.e., either by updating or re-training the data-driven model [40,41,42]). It can learn dynamic features to adapt to the trends of the new input time series data. This makes DIL another potential solution to the shortage of available data for FTC tasks. However, DIL in the field of FTC has undergone limited investigation. The quantity and quality of available modeling data are important factors that affect the accuracy of FTC models. FTC models using the DIL strategy can be updated or retrained using newly collected data from an HVAC system. By learning from recent data, the adaptability and calibration accuracy of the FTC model can be improved. Many data-driven sensor calibration methods have been investigated for different amounts of training data [43]. Ng et al. [40] proposed an enhanced self-organizing incremental neural network for evaluating the potential of incremental learning. In a machine-learning-based strategy for building energy prediction, Singh et al. [41] reduced the prediction error via data incremental learning and enrichment.

The application of DIL in FTC tasks for HVAC systems is relatively understudied. There is a lack of sufficient investigations on the influence of DIL on FTC models’ performance in new building energy system scenarios with increasing amounts of data. In addition, it is unclear whether DIL is applicable for improving FTC performance. Answers to these questions would substantially contribute to the selection of appropriate FTC solutions for real-world applications and improve performance.

### 1.4. Research Objectives

To address the current research gap, this study proposed an in situ selective incremental calibration strategy. Multiple linear regression–Bayesian inference and principal component analysis were used to realize in situ sensor calibration and data filtering, respectively. The proposed data incremental strategy not only used historical datasets to update the calibration model but also retrained the model using all the data, including the newly generated data from the building energy system. Based on the EnergyPlus–Python co-simulation testbed, an FTC study was carried out by selecting three target sensors in a central air-conditioning (CAC) system. Finally, the impacts of the data quality (under noise and steady-state conditions), data volume, and number of variables were evaluated. The research objectives of this study can be summarized as follows:(1)The first objective was to propose an FTC strategy for the in situ selective incremental calibration of HVAC system sensors based on the multiple linear regression–Bayesian inference (MLR-BI) and principal component analysis (PCA) methods.(2)The second objective was the quantification of the FTC results for three target sensors in the central air-conditioning system, including the target variables, energy consumption, and thermal comfort.(3)The last objective was to explore the impacts of several influencing factors (i.e., the data quality, data volume, and number of variables) on the FTC results and determine the appropriate FTC strategies for the target sensors.

## 2. Principle of In Situ Selective Incremental Calibration

This study proposed an ISIC strategy. The strategy consists of multiple linear regression–Bayesian inference and principal component analysis. These two methods are used to realize in situ sensor calibration and data filtering, respectively. Principal component analysis can effectively filter the data samples for calibration and reduce the influence of outliers on the calibration results. Then, multiple linear regression–Bayesian inference employs these filtered data samples for in situ sensor calibration, providing more accurate calibrated data. Ultimately, these calibrated data are fed into the CAC system for FTC. These two methods are presented in Section 2.1 and Section 2.2.

### 2.1. Fault Calibration Using Multiple Linear Regression–Bayesian Inference

In the sensor calibration, Bayesian inference is applied to calculate the offset constants and unknown parameters in the model to minimize the distance function [44]. As shown in Equations (1)–(3), the prior probability function of the error x is denoted as π(x), and the distance function $D\left(x\right)$ is constructed to derive the likelihood function ($P\left(Y_{b}|x\right)$). The normalization function ($P\left(Y_{b}\right)$) of the fault values is obtained by sampling with the Markov chain Monte Carlo algorithm [45]. Finally, the posterior distribution ($P\left(x|Y_{b}\right)$) and the error x are determined. In addition, σ denotes the standard deviation of the prior distribution.
(1)$$P\left(Y_{b}|x\right)=\frac{1}{\delta \sqrt{2\pi }}\exp\left[-\frac{1}{2\delta^{2}}D\left(x\right)\right]$$
(2)$$P\left(Y_{b}\right)=\int P\left(Y_{b}|x\right)\pi \left(x\right)dx$$
(3)$$P\left(x|Y_{b}\right)=\frac{P\left(Y_{b}|x\right)\times \pi \left(x\right)}{P\left(Y_{b}\right)}$$

As shown in Equation (4), multiple linear regression (MLR) is used in this study to construct the system term [46,47] and incorporate the sensor-level term to improve the calibration accuracy [29]. $f(V_{c}^{\prime })$ in Equation (6) represents the regression function of the system-level model containing the target sensor to be calibrated. $f\left(V_{p}\right)$ in Equation (5) is the benchmark of the system and represents all variable information in the system except the target sensor to be calibrated. The compensation function for the target sensor is defined as Equation (7).
(4)$$D\left(x\right)=\underset{system\;term}{\underbrace{\sum_{p}^{P}\sum_{c}^{C}{\left(f\left(V_{p}\right)-f(V_{c}^{\prime })\right)}^{2}}}+\underset{sensor\;term}{\underbrace{\sum_{c}^{C}\left(V_{B,c}-V_{c}^{\prime }\right)}}$$
(5)$$f\left(V_{p}\right)=\alpha_{1}V_{1}+\alpha_{2}V_{2}+\ldots +\alpha_{P}V_{P}+\alpha_{0}$$
(6)$$f\left(V_{c}^{\prime }\right)=\beta_{1}V_{1}^{\prime }+\beta_{2}V_{2}^{\prime }+\ldots +\beta_{c}V_{c}^{\prime }+\beta_{0}$$
(7)$$V_{c}^{\prime }=V_{f}+x$$
where $V_{1}^{\prime }-V_{c}^{\prime }$ is the target sensor to be calibrated, $V_{1}-V_{p}$ is the physical sensor other than the target sensor to be calibrated, $\alpha_{0}\;and\;\beta_{0}$ are the constant terms of the MLR model, and $\alpha_{1}-\alpha_{p}$ and $\beta_{1}-\beta_{c}$ are the coefficients corresponding to each of the above variables. $V_{B,c}$ is the benchmark of the target sensor to be calibrated. $V_{f}$ is the fault data.

### 2.2. Data Filtering Using Principal Component Analysis

Principal component analysis (PCA) is a commonly used method for sensor fault detection and process-monitoring methods in industry areas [48]. For a given multivariate dataset X, PCA transforms a set of correlated original variables into a new set of mutually uncorrelated or orthogonal forms. In HVAC systems, individual sensors are often multidimensional and interrelated. Therefore, the original variables can be represented with fewer principal components due to the redundancy between the variables. During the data transformation process, each data sample is decomposed into two subspaces, the principal component space and the residual subspace [49]. For any new test data sample z, the original data are divided into two parts, as shown in Equation (8):(8)$$z=\hat{z}+\tilde{z}$$
where $\hat{z}$ and $\tilde{z}$ denote the projections in the principal component subspace and residual subspace, respectively. Typically, $\tilde{z}$ is used to reflect anomalous deviations from the original statistical correlations between variable measurements. As shown in Equation (9), the Q statistic is built in the residual subspace to detect anomalous deviations, such as sensor bias.
(9)$$Q_{z}=\left\|\tilde{z}\right\|={\tilde{z}}^{T}\tilde{P}{\tilde{P}}^{T}\tilde{z}\leq Q_{a}$$
where $Q_{a}$ is the computational threshold for the Q statistic $Q_{z}$ used to detect sensor faults. In this study, the principal element selection method of cumulative contribution was used to select the number of principal components and thus determine the threshold $Q_{a}$. This method defines the number of principal elements as k when the cumulative contribution of the kth principal element is greater than or equal to some empirical value. A cumulative contribution of variation greater than or equal to 85% is used as the basis for judging the selection of the number of principal elements [50]. The cumulative contribution is shown in Equation (10), and $Q_{a}$ can be found using Equation (11).
(10)$${CV}_{k}=\sum_{i=1}^{k}\lambda_{i}/\sum_{i=1}^{n}\lambda_{i}$$
(11)$$Q_{a}=\theta_{1}{\left[\frac{c_{\alpha }\sqrt{2\theta_{2}h_{0}^{2}}}{\theta_{1}}+1+\frac{\theta_{2}h_{0}(h_{0}-1)}{\theta_{1}^{2}}\right]}^{\frac{1}{h_{0}}}$$
where $c_{\alpha }$ is the normal deviation corresponding to the upper (1−α) percentile, $\theta_{i}=\sum_{j=k+1}^{n}\lambda_{j}^{i}$, $h_{0}=1-2\theta_{1}\theta_{3}/3\theta_{2}^{2}$, and $\lambda_{i}$ is the ith eigenvalue of the covariance array. If the Q statistic is greater than its corresponding threshold $Q_{a}$, there is a sensor fault. If the Q statistic does not exceed $Q_{a}$, there is no sensor fault.

## 3. Research Methodology

### 3.1. Research Framework

Figure 1 illustrates the research framework of this study, including three main steps:(1)The FTC strategy for in situ selective incremental calibration (ISIC) was proposed. MLR-BI and PCA were used to realize fault calibration and data filtering, respectively. A brief flow of the ISIC strategy is shown in Appendix A.(2)The fault modeling and FTC of the indoor air thermostat (*T*_tz1_), supply air temperature (*T_sa_*), and chilled water supply temperature (*T_chws_*) in the CAC system using the EnergyPlus–Python co-simulation testbed was carried out to demonstrate the variations in target variables and energy consumption on a typical day and the changes in thermal comfort in August.(3)The effects of the data quality, data volume, and number of variables on the FTC results were evaluated.
Figure 1Research framework of this study.
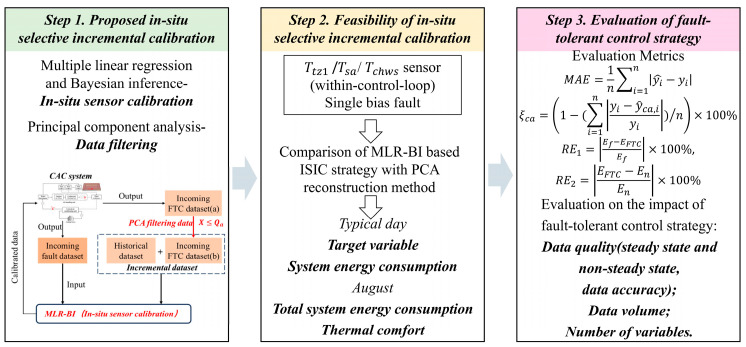



### 3.2. Process of ISIC Fault-Tolerant Control Strategy

In this study, an FTC strategy for in situ selective incremental calibration was proposed based on an in situ calibration (IC) strategy (Figure 2a). Figure 2b shows the in situ selective incremental calibration process in detail. At $\tau_{1}$, the calibrated data $T_{c,1}$ are obtained from the fault dataset ($\tau_{1}$) and the historical dataset via MLR-BI calculation. The sample $X_{\tau_{1}}$ can be obtained by inputting the calibrated data into the HVAC system. At the moment $\tau_{2}$, the sample $X_{\tau_{1}}$ is first filtered with the threshold $Q_{a}$ calculated using PCA. If $X_{\tau_{1}}$ is less than or equal to $Q_{a}$, then $X_{\tau_{1}}$ and the historical dataset constitute the incremental dataset ($\tau_{2}$). The incremental dataset ($\tau_{2}$) and the fault dataset ($\tau_{2}$) are passed through MLR-BI to obtain the calibrated data $T_{c,2}$. These data are carried over to the HVAC system to obtain the sample $X_{\tau_{2}}$, and so on, until the end of the simulation. The incremental dataset size is shown in Equation (12):(12)$${Size}_{ISIC}=(m+\left(\frac{t_{PCA\left(X\leq Q_{a}\right)}}{∆\tau }\right))\times n$$
where τ denotes the time of calibration; ∆τ denotes the time step; X denotes the dataset, which is of size *m* × *n*; and $Q_{a}$ is the threshold calculated via PCA from the historical dataset.

### 3.3. Thermal Comfort Metrics

In this study, the predicted mean valuation (PMV) and predicted percentage dissatisfaction (PPD) were selected to evaluate indoor thermal comfort. The PMV model was proposed by Fanger in the late 1960s [51]. The PPD represents the percentage of the population that is dissatisfied with the thermal environment. As shown in Table 1, the PMV represents the average of hot and cold sensations in the majority of people in the same environment, and the metric utilizes a 7-level scale.

### 3.4. Evaluation of FTC Strategy

#### 3.4.1. Evaluation Metrics

The target variable data under FTC conditions were compared with the target variable data under normal conditions to calculate the mean absolute error (MAE), as shown in Equation (13). The smaller the MAE, the higher the accuracy of the FTC data. In addition, this paper used the calibration accuracy ($\xi_{ca}$) to measure the results of the calibration method, as shown in Equation (14).
(13)$$MAE=\frac{1}{n}\sum_{i=1}^{n}\left|\hat{y_{i}}-y_{i}\right|$$
(14)$$\xi_{ca}=\left(1-\frac{\sum_{i=1}^{n}\left|\frac{y_{i}-{\hat{y}}_{ca,i}}{y_{i}}\right|}{n}\right)\times 100\%$$
where $\hat{y_{i}}$ is the target variable data for the FTC, $y_{i}$ is the target variable data of the CAC system under normal conditions, and ${\hat{y}}_{ca,i}$ is the calibrated data of the target variable. As shown in Equations (15) and (16), the ${RE}_{f}$ was used to quantify the degree of energy consumption deviation from the fault energy consumption after FTC. The ${RE}_{n}$ was used to quantify the degree of energy consumption close to normal levels after FTC.
(15)$${RE}_{f}=\left|\frac{E_{f}-E_{FTC}}{E_{f}}\right|\times 100\%$$
(16)$${RE}_{n}=\left|\frac{E_{FTC}-E_{n}}{E_{n}}\right|\times 100\%$$
where $E_{f}$, $E_{n}$, and $E_{FTC}$ are the total energy consumption for the CAC system’s operation under fault, normal, and FTC conditions, respectively.

#### 3.4.2. Analysis of the Influencing Factors of MLR-BI-Based FTC Strategies

This study considered two FTC strategies, in situ calibration and in situ selective incremental calibration, applied in a real operating environment and evaluated in terms of the following aspects:(1)Data quality: The results of the FTC of the two strategies were evaluated using steady-state (10:30–17:30) versus non-steady-state data and different standard deviations of noise (SDN) (0.01, 0.05, 0.1, 0.15, 0.5, 1, 1.5, and 2.0), respectively.(2)Data volume: Different time periods of 1 day, 7 days, 14 days, 1 month, 2 months, and 3 months were selected as the original training set for MLR-BI.(3)The number of variables: Regression models for the target sensors in different variable scenarios were constructed as shown in Figure 3.

## 4. Case Study

### 4.1. Central Air-Conditioning System and Target Building

Figure 4 shows the schematic diagram of the target CAC system, which was a CAC system consisting of an independent fresh air system (air-handling units) and an indoor terminal (a fan coil). The system design was based on the official EnergyPlus example (5ZoneFanCoilDOASCool) [52]. The target building was located in Chicago, IL, USA, and was a single-story office building. This building had a core thermal zone in the middle and four thermal zones around it. The building had a floor area of 465 m^2^, a height of 3.0 m, and a return air chamber at the top (with a height of 0.6 m). Each zone had a return air chamber and windows, and the cooling (heat) loads of the five thermal zones were shared by an independent fresh air system and five indoor terminals.

### 4.2. Operation Schedule and Fault Settings

Table 2 depicts the operating schedule of the target system. The CAC system was switched on at 07:00, and a bias fault was introduced into the target sensor at 12:00. The FTC method using data increments was used to calibrate the system starting at 12:30, which, in turn, achieved FTC. In addition, the system operated from 07:00 to 18:00 on weekdays. The system simulation model data output time step was 30 min. In this study, three within-control-loop sensors were selected, which were the thermostat of hot zone 1, the supply air temperature of the air-handling unit, and the chilled water supply temperature sensor of the CAC mainframe. Table 3 describes the setpoints of the target sensors and the magnitudes of the faults introduced.

## 5. Results and Discussion

### 5.1. PCA Filtering Fault-Tolerant Data Results

In this section, the $T_{tz1}$ thermostat is used as an example to show the PCA filtering data. The training set was data from July during the operation of the CAC system. The confidence level was 0.05. Figure 5 shows the $T_{tz1}$ thermostat filtering data using the PCA fault detection algorithm in August. The number of principal components was three and the cumulative contribution was 91.12%. The threshold was 1.64. Samples with Q-statistics exceeding the threshold were rejected and did not participate in the subsequent in situ incremental calibration.

### 5.2. Comparison of the Accuracy between MLR-BI Calibration and PCA Reconstruction

Via the ISIC strategy, the PCA reconstruction method was selected in this section for a comparison study with the MLR-BI calibration method. Figure 6 shows the calibration accuracy of the target sensors for both the steady-state and non-steady-state data cases. In most situations, the calibration accuracy of the MLR-BI method was better than that of PCA reconstruction. For the $T_{tz1}$ thermostat and $T_{sa}$ sensor, the calibration accuracy using MLR-BI was greater than that using PCA reconstruction in the steady-state case. For the $T_{tz1}$ thermostat and $T_{sa}$ sensor, the MLR-BI calibration accuracies were 97.4% and 96.6%, respectively. For the $T_{tz1}$ thermostat, the calibration accuracy using MLR-BI was greater than that using PCA reconstruction in the non-steady-state case. The $T_{tz1}$ thermostat was calibrated to 97.9% and 94.4% accuracies using MLR-BI and PCA, respectively. The $T_{chws}$ sensor was calibrated to almost the same accuracy using both methods.

### 5.3. FTC Results of CAC Temperature Sensor Bias Faults

#### 5.3.1. FTC Results for a Typical Day

Figure 7 shows the changes in the temperature and energy consumption of the CAC system on a typical day (1 August) under different conditions. After the fault was introduced into the sensor, the measurement data showed a fault deviation. This resulted in a deviation in the original set temperature from the normal temperature. When the measurement data were calibrated, the set temperature was allowed to return to its original state. At 12:00 p.m., a +2 °C bias fault was introduced into the $T_{tz1}$ thermostat. The $T_{tz1}$ dropped from 26.00 °C to 24.86 °C. At 13:00 p.m., either the IC or ISIC method was used to bring the temperature back up to 25.64 °C, and the energy consumption decreased from 14.88 M to 14.24 MJ. When a +2 °C bias fault was introduced into the $T_{sa}$ sensor, the $T_{sa}$ decreased from 14.00 °C to 12 °C, and the energy consumption increased from 15.03 MJ to 15.20 MJ. When FTC was performed at 13:00, the $T_{sa}$ and energy consumption were 13.90 °C and 14.09 MJ, respectively. At 13:00, the $T_{chws}$ increased from 5.00 °C to 7.00 °C, and the energy consumption was reduced from 15.60 MJ to 13.87 MJ.

#### 5.3.2. Thermal Comfort in August

Figure 8 depicts the average PPD and PMV in August in thermal zone 1. In the three sensors, the $T_{tz1}$ thermostat fault had a greater impact on indoor thermal comfort. After FTC with IC and ISIC, the PPD decreased from 10.13% to 9.45% and 9.46%, respectively. The PMV increased from −0.38 to −0.07 and −0.06, respectively. Indoor dissatisfaction was reduced, and thermal sensation was more moderate. For the $T_{sa}$ sensor after FTC with IC and ISIC, the PPD decreased from 10.20% to 9.44% and 9.57%, respectively. The PMV increased from −0.12 to −0.05 and −0.06, respectively. The thermal environment in thermal zone 1 was restored to normal levels.

### 5.4. Factors Impacting FTC Results

#### 5.4.1. Data Quality

(a)Data Noise

For the target sensors, Figure 9 and Figure 10 show the FTC results in August under different standard deviation of noise (SDN) conditions, respectively. Oriented to different SDNs, both the IC and ISIC strategies could maintain a good calibration accuracy. Using these two strategies, the MAE of the $T_{tz1}$ thermostat varied less, at approximately 0.17 °C. The ${RE}_{f}$ and ${RE}_{n}$ were approximately 3.00% and 0.20%, respectively. In this case, the FTC results with ISIC were better than those with IC. When the SDN was from 0.01 to 0.15, the MAE of $T_{sa}$ was small and stable, at less than 0.02 °C. The ${RE}_{f}$ was higher than 4.60%, and the ${RE}_{n}$ was lower than 0.05%. When the SDN was between 0.01 and 0.15, the FTC results with IC and ISIC were almost the same. Except for the $T_{chws}$ sensor at SDN = 0.1, the FTC results for both strategies were better at SDN. At other standard deviations of noise, the MAE was lower than 0.1 °C, the ${RE}_{f}$ was close to 5.00%, and the ${RE}_{n}$ was lower than 0.05%. At SDN = 2, the ${RE}_{f}$ and ${RE}_{n}$ of ISIC were 4.8% and 0.1%, respectively, which are better than the FTC results with IC. The ISIC strategy excluded the calibrated data that exceeded the threshold during the data increment process, which improved the calibration accuracy and made the FTC results better.

(b)Steady State and Non-steady State

Table 4 and Table 5 show the FTC results for the target sensors in a steady state and non-steady state. For the $T_{tz1}$ thermostat, the FTC results were better with the IC strategy using steady-state data and the ISIC strategy using non-steady-state data. The FTC results with ISIC were better than those with IC when non-steady-state data were used. For the $T_{tz1}$ thermostat, the MAE, ${RE}_{f}$, and ${RE}_{n}$ were 0.170 °C, 2.98%, and 0.20% for the FTC with ISIC using non-steady-state data, respectively. When steady-state data were used, the FTC results with IC were better than the results with ISIC for the $T_{tz1}$ thermostat. In this situation, the MAE, ${RE}_{f}$, and ${RE}_{n}$ were 0.168 °C, 3.39%, and 0.22%, respectively. For the $T_{sa}$ sensor, the FTC results with IC were better and independent of the steady-state and non-steady-state data. In the case of the ISIC strategy, the FTC results were better using steady-state data for the $T_{sa}$ sensor. The MAE, ${RE}_{f}$, and ${RE}_{n}$ were 0.105 °C, 4.41%, and 0.26%, respectively. The calibration accuracy of the $T_{chws}$ sensor was almost the same with the IC and ISIC strategies whether steady-state data or non-steady-state data were used.

#### 5.4.2. Data Volume

For the target sensors, Figure 11 and Figure 12 show the FTC results with IC and ISIC for different data volumes. The FTC results for a 1-day volume were better when the $T_{tz1}$ thermostat adopted the IC strategy. In this situation, the MAE, ${RE}_{f}$, and ${RE}_{n}$ were 0.15 °C, 3.30%, and 0.13%, respectively. When the data volume was in the range of 7 days to 3 months, the FTC results of IC and ISIC were almost the same and deteriorated with the increase in the data volume. When 1-month data were selected for the $T_{sa}$ sensor, the FTC results with IC were better, with the MAE, ${RE}_{f}$, and ${RE}_{n}$ being 0.07 °C, 4.49%, and 0.18%, respectively. When other data amounts were selected, the FTC results with IC and ISIC were almost the same. For the $T_{chws}$ sensor, the calibration accuracies of IC and ISIC were close to each other when the amount of data was small. When a 3-month volume was chosen, the MAE, ${RE}_{f}$, and ${RE}_{n}$ of IC were 0.20 °C, 4.51%, and 0.04%, respectively. The results of IC were better than those of ISIC. Low outside temperatures exist in May and June. This could result in the possible shutdown of the CAC system. The data quality was not good in May and June. This also caused poor FTC results for the target sensor using 2-month (June–July) and 3-month (May–July) data.

#### 5.4.3. Variable Number

Figure 13 and Figure 14 show the FTC results with IC and ISIC in different variable scenarios for the target sensors. The FTC results of the $T_{tz1}$ sensor using IC and ISIC were relatively similar. Both strategies achieved better FTC results in variable scenario A, and the MAE, ${RE}_{f}$, and ${RE}_{n}$ were 0.17 °C, 2.98%, and 0.20% when ISIC was used, respectively. In different variable scenarios, the FTC results of the $T_{sa}$ sensor were better in variable scenario F. The FTC results with ISIC were slightly better than those with IC, and the MAE and ${RE}_{f}$ were 0.17 °C and 4.65%, respectively. For different variable scenarios, the FTC results of the $T_{chws}$ sensor with IC and ISIC were basically the same, and the ${RE}_{f}$ was approximately 4.87%.

## 6. Conclusions

This paper proposed an in situ selective incremental calibration strategy. This strategy was used to address the problem of the lack of available information about HVAC systems affecting the performance of data-driven fault-tolerant control models. Using the EnergyPlus–Python co-simulation testbed, the central air-conditioning system of a single-story office building was simulated, and faults were introduced into the $T_{tz1}$ thermostat, $T_{sa}$ sensor, and $T_{chws}$ sensor. This study quantified the changes in target variables, energy consumption, and thermal comfort before and after fault-tolerant control. The effects of the data quality, data volume, and number of variables on the fault-tolerant control results were evaluated. The main conclusions are as follows:
(1)The fault-tolerant control strategy using data increments can lead to good fault-tolerant control results for a central air-conditioning system. Compared with the sensor fault operation, the fault-tolerant control strategy reduced the total energy consumption by 2.98%, 3.72%, and 4.87% for the $T_{tz1}$ thermostat and the faulty $T_{sa}$ and $T_{chws}$ sensors, respectively. For the $T_{tz1}$ thermostat and faulty $T_{sa}$, the predicted percentage dissatisfaction was reduced by 0.67% and 0.63%, respectively. The system energy consumption and indoor thermal comfort were close to normal levels after fault-tolerant control.(2)For the $T_{tz1}$ thermostat and $T_{sa}$ sensor, better fault-tolerant control results were obtained by using in situ selective incremental calibration when the standard deviation of noise was small. When non-steady-state data were used, better results were obtained by using in situ selective incremental calibration for the $T_{tz1}$ thermostat. For the $T_{chws}$ sensor, the data quality had less influence on the fault-tolerant control results.(3)Compared with in situ calibration, the $T_{tz1}$ thermostat obtained good fault-tolerant control results with the in situ selective incremental calibration strategy with a 7-day data volume and sufficiently variable scenarios. The $T_{sa}$ and $T_{chws}$ sensors obtained better fault-tolerant control results with the in situ selective incremental calibration strategy with a 14-day data volume and variable scenarios with limited information.


## Figures and Tables

**Figure 2 sensors-24-01150-f002:**
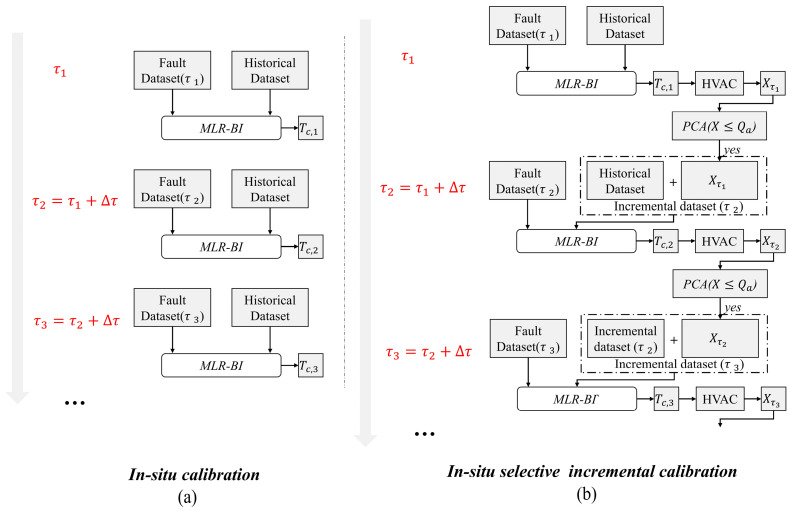
Flow chart of in situ calibration and in situ selective incremental calibration.

**Figure 3 sensors-24-01150-f003:**
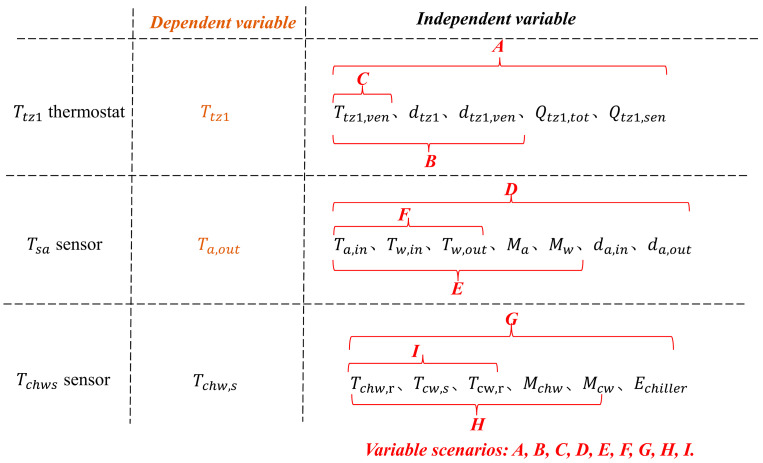
Target sensor variable scenario settings. Note: Taking the variable scenario G as an example, the dependent variable involved in the MLR-BI modeling is the chilled water supply temperature. The independent variables are the chilled water return temperature ($T_{chw,r}$), the cooling water supply temperature ($T_{cw,s}$), the cooling water return temperature ($T_{cw,r}$), the chilled water mass flow rate ($M_{chw}$), the cooling water mass flow rate ($M_{cw}$), and the chiller’s energy consumption ($E_{chiller}$).

**Figure 4 sensors-24-01150-f004:**
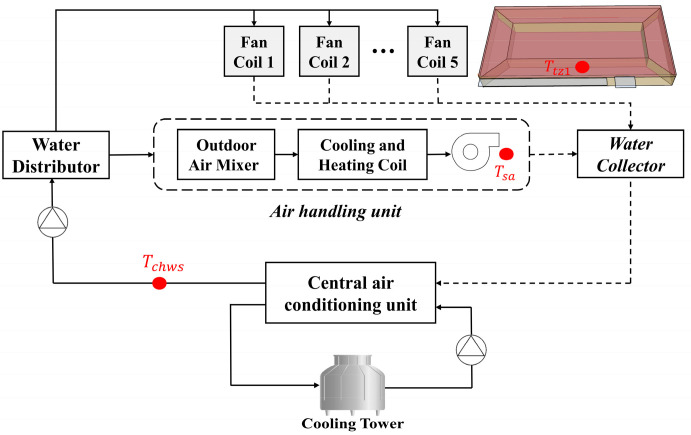
Schematic diagram of CAC system.

**Figure 5 sensors-24-01150-f005:**
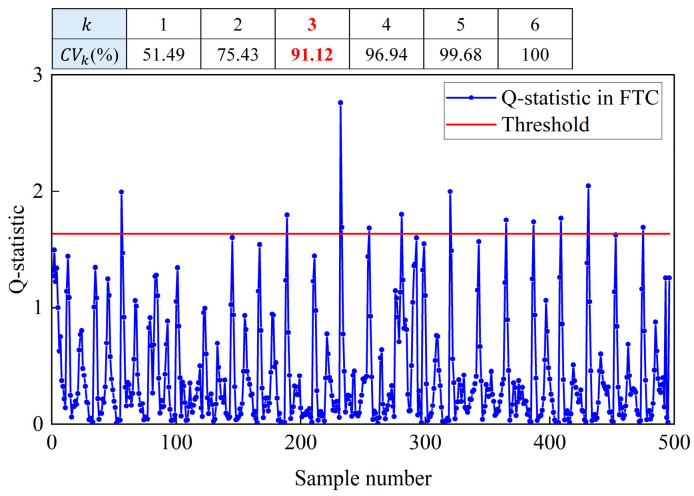
Demonstration of PCA filtering of fault-tolerant data using $T_{tz1}$ thermostat as an example. Note: The cumulative contribution was 91.12% and greater than 85%. Then the number of principal components was three.

**Figure 6 sensors-24-01150-f006:**
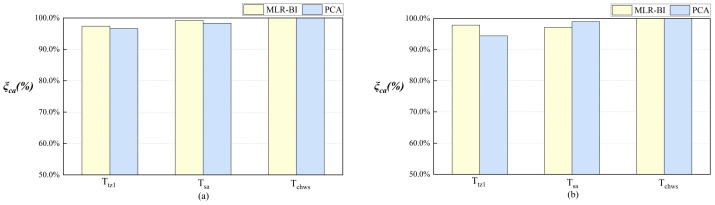
Comparison of accuracy using MLR-BI calibration and PCA reconstruction. (**a**) Steady-state data. (**b**) Non-steady-state data.

**Figure 7 sensors-24-01150-f007:**
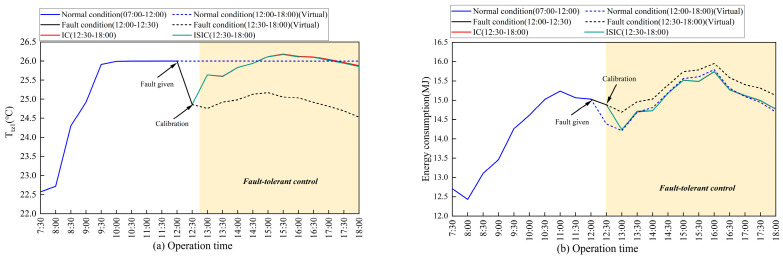
Changes in target variables and system energy consumption under three operating conditions on 1 August: (**a**) $T_{tz1}$; (**b**) system energy consumption for $T_{tz1}$ thermostat; (**c**) $T_{sa}$; (**d**) system energy consumption for the $T_{sa}$ sensor; (**e**) $T_{chws}$; and (**f**) system energy consumption of the $T_{chws}$ sensor.

**Figure 8 sensors-24-01150-f008:**
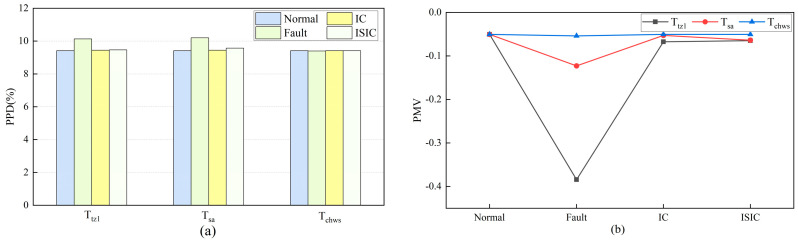
Changes in thermal comfort before and after FTC in thermal zone 1 in August: (**a**) PPD; (**b**) PMV.

**Figure 9 sensors-24-01150-f009:**
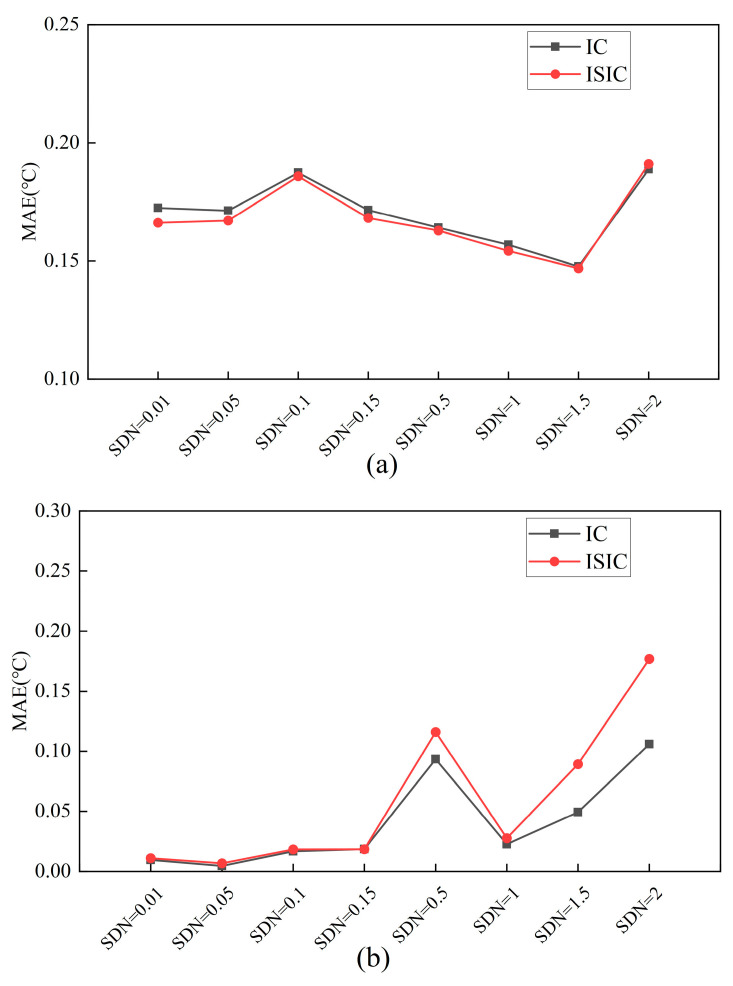
MAEs of target sensors for IC and ISIC FTC with different SDNs: (**a**) $T_{tz1}$ thermostat; (**b**) $T_{sa}$ sensor; and (**c**) $T_{chws}$ sensor.

**Figure 10 sensors-24-01150-f010:**
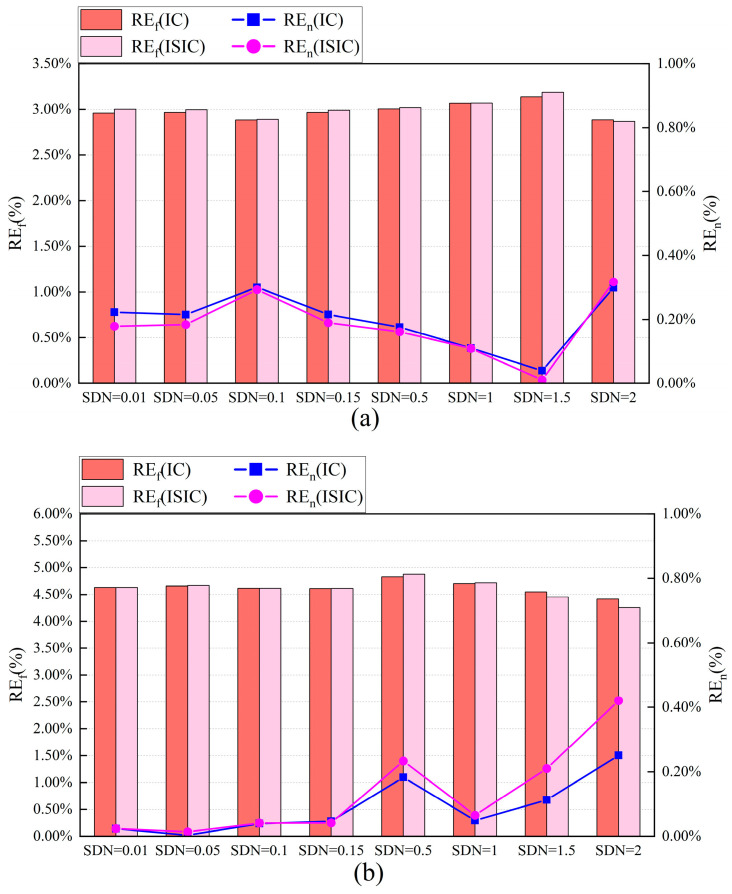
Relative errors of August energy consumption of target sensors for IC and ISIC FTC with different SDNs: (**a**) $T_{tz1}$ thermostat; (**b**) $T_{sa}$ sensor; and (**c**) $T_{chws}$ sensor.

**Figure 11 sensors-24-01150-f011:**
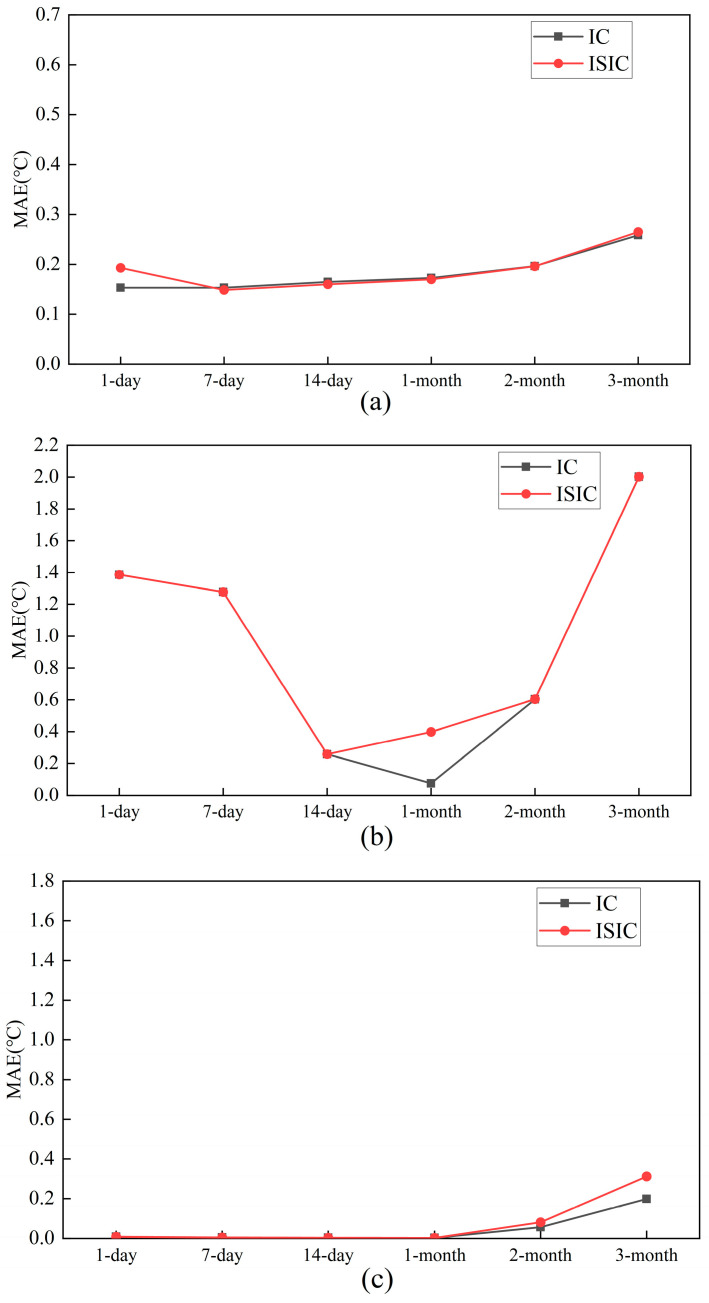
MAEs of target sensors for IC and ISIC with different data volumes: (**a**) $T_{tz1}$ thermostat; (**b**) $T_{sa}$ sensor; and (**c**) $T_{chws}$ sensor.

**Figure 12 sensors-24-01150-f012:**
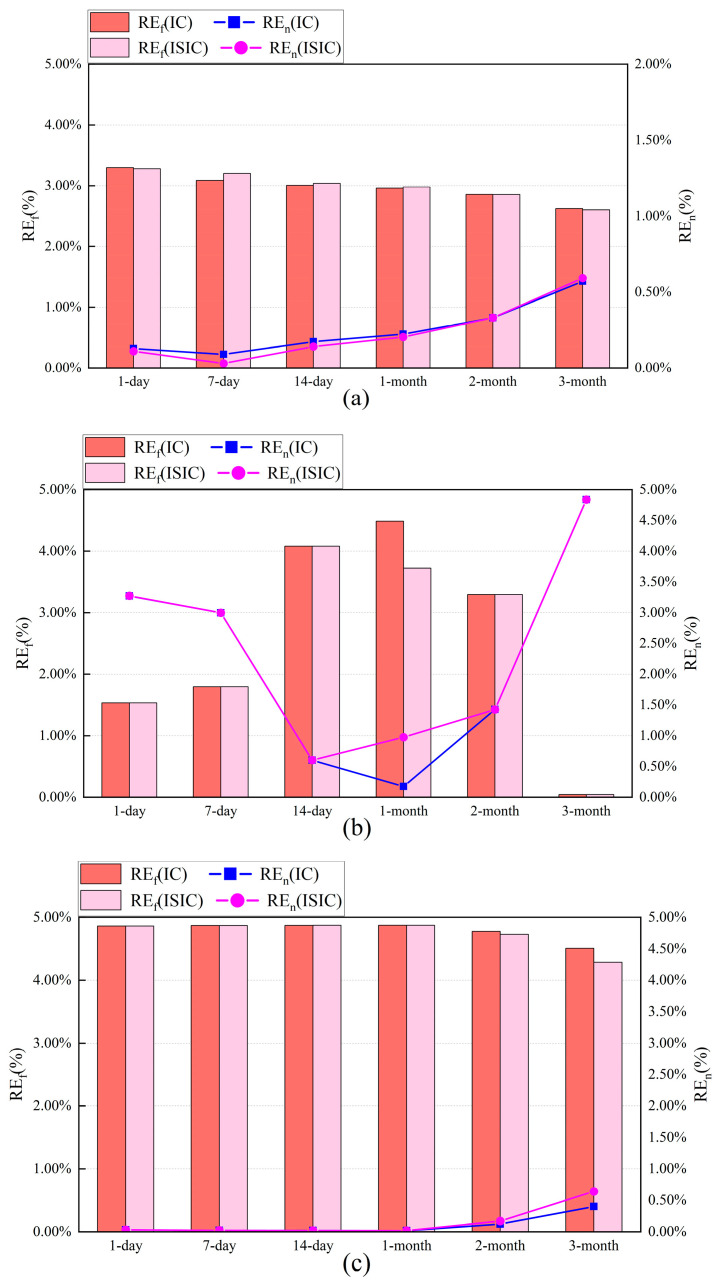
Relative errors of August energy consumption of target sensors for IC and ISIC with different data volumes: (**a**) $T_{tz1}$ thermostat; (**b**) $T_{sa}$ sensor; and (**c**) $T_{chws}$ sensor.

**Figure 13 sensors-24-01150-f013:**
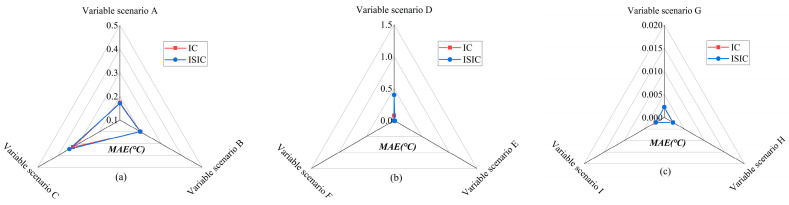
MAEs of target sensors for IC and ISIC in different variable scenarios: (**a**) $T_{tz1}$ thermostat; (**b**) $T_{sa}$ sensor; and (**c**) $T_{chws}$ sensor.

**Figure 14 sensors-24-01150-f014:**
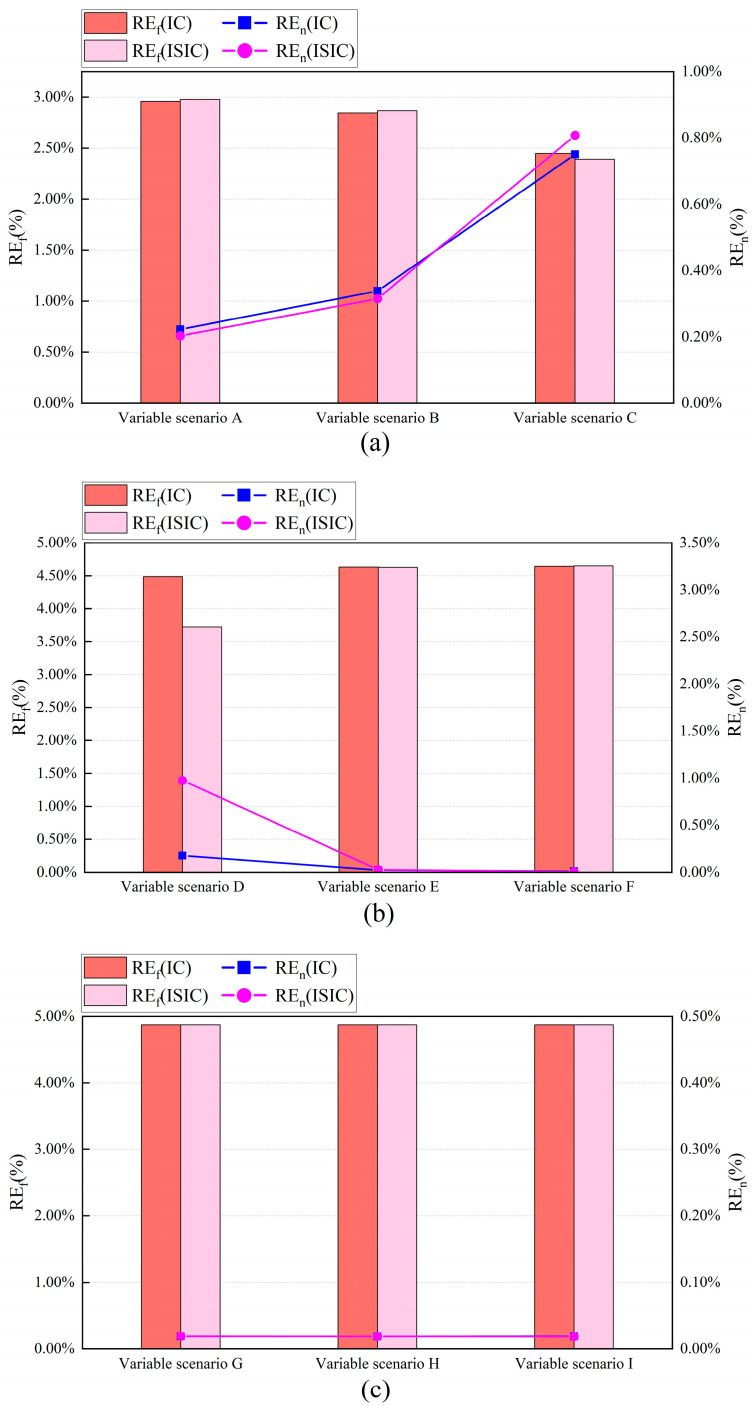
Relative errors of energy consumption in August of target sensors for IC and ISIC in different variable scenarios: (**a**) $T_{tz1}$ thermostat; (**b**) $T_{sa}$ sensor; and (**c**) $T_{chws}$ sensor.

**Table 1 sensors-24-01150-t001:** PMV thermal sense scale.

Thermal Sensation	Hot	Warm	Mildly Warm	Moderate	A Little Cool	Cool	Cold
PMV	+3	+2	+1	0	−1	−2	−3

**Table 2 sensors-24-01150-t002:** CAC system operation schedule.

Period	Conditions
07:00–12:00	Normal
12:00–12:30	Fault
12:30	Start of in situ calibration
12:30–18:00	FTC

**Table 3 sensors-24-01150-t003:** Target sensor setpoints and fault settings.

Target Sensor	Setpoint	Bias Amplitude
$T_{tz1}$	26 °C	+2 °C
$T_{sa}$	14 °C	+2 °C
$T_{chws}$	7 °C	+2 °C

**Table 4 sensors-24-01150-t004:** MAEs of target sensors during IC and ISIC in steady state and non-steady state.

Target Sensor	IC		ISIC	
	Non-Steady State	Steady State	Non-Steady State	Steady State
$T_{tz1}$	0.173 °C	0.168 °C	0.170 °C	0.176 °C
$T_{sa}$	0.075 °C	0.075 °C	0.399 °C	0.105 °C
$T_{chws}$	0.002 °C	0.003 °C	0.002 °C	0.003 °C

**Table 5 sensors-24-01150-t005:** Relative errors of August energy consumption of target sensors during IC and ISIC in steady state and non-steady state.

Target Sensor	Data State	IC		ISIC	
		*RE_f_*	*RE_n_*	*RE_f_*	*RE_n_*
$T_{tz1}$	Non-steady state	2.96%	0.22%	2.98%	0.20%
	Steady state	3.39%	0.22%	3.42%	0.26%
$T_{sa}$	Non-steady state	4.49%	0.18%	3.72%	0.98%
	Steady state	4.48%	0.18%	4.41%	0.26%
$T_{chws}$	Non-steady state	4.87%	0.02%	4.87%	0.02%
	Steady state	4.87%	0.02%	4.87%	0.02%

## Data Availability

Data are contained within the article.

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
