# Peer review of "Study on Sensor Fault-Tolerant Control for Central Air-Conditioning Systems Using Bayesian Inference with Data Increments"

_sensors, 2024, doi:10.3390/s24041150_

Round 1
Reviewer 1 Report
Comments and Suggestions for Authors
This paper presents an in-situ selective incremental calibration method for the data-driven fault-tolerant control in central air conditioning systems. The paper conducts a detailed literature review, and presents the proposed method and the evaluation results clearly. In general the paper is in a good shape.
A minor suggestion: in both abstract and conclusion, instead of showing “reduced to”, consider showing “reduced from XXX to YYY” or simply showing “reduced by ZZZ%”. This gives readers a clear understanding of the benefits achieved by your method.
Comments on the Quality of English Language
In general, this paper's quality of English is good.
Reviewer 2 Report
Comments and Suggestions for Authors
(1) In the manuscript, PPD stands for Predicted Percentage Dissatisfaction, but this is not explained in the abstract where the abbreviation appears first.
(2) In the case study, the choice of Ttz1, T sa, T chws as experimental sensors over other sensors needs further explanation.
(3) Table 2 shows a calibration time of 12:10, which contradicts the calibration time of 12:30 in Figure 5.
(4) Whether in SDN or data volume experiments, in some specific cases, the results of IC are superior to international standard industrial classifications. Is there any additional explanation for these? Different experimental phenomena may explain these results.

Comments on the Quality of English Language
Appropriate English corrections are required.
Reviewer 3 Report
Comments and Suggestions for Authors
Please see the attachment.

Comments on the Quality of English Language
Moderate editing of English language required.
Round 2
Reviewer 3 Report
Comments and Suggestions for Authors
All the recommendations and suggestions have been incorporated by the authors. Therefore, I recommend this manuscript for acception.
Comments on the Quality of English Language
English language and style are fine.